# Generation of third-harmonic spin oscillation from strong spin precession induced by terahertz magnetic near fields

Zhenya Zhang[1], Fumiya Sekiguchi [1], Takahiro Moriyama [1], Shunsuke C. Furuya[2], Masahiro Sato[3], Takuya Satoh [4], Yu Mukai[5], Koichiro Tanaka [6], Takafumi Yamamoto [7], Hiroshi Kageyama [8], Yoshihiko Kanemitsu [1] ✉ & Hideki Hirori [1] ✉

The ability to drive a spin system to state far from the equilibrium is indispensable for investigating spin structures of antiferromagnets and their functional nonlinearities for spintronics. While optical methods have been considered for spin excitation, terahertz (THz) pulses appear to be a more convenient means of direct spin excitation without requiring coupling between spins and orbitals or phonons. However, room-temperature responses are usually limited to small deviations from the equilibrium state because of the relatively weak THz magnetic fields in common approaches. Here, we studied the magnetization dynamics in a $HoFeO_3$ crystal at room temperature. A custom-made spiral-shaped microstructure was used to locally generate a strong multicycle THz magnetic near field perpendicular to the crystal surface; the maximum magnetic field amplitude of about 2 T was achieved. The observed time-resolved change in the Faraday ellipticity clearly showed second- and third-order harmonics of the magnetization oscillation and an asymmetric oscillation behaviour. Not only the ferromagnetic vector **M** but also the antiferromagnetic vector **L** plays an important role in the nonlinear dynamics of spin systems far from equilibrium.

Nonlinear responses can be used to probe and control quantum states of matter in strong electromagnetic fields. Therefore, charge and phonon degrees of freedom have been intensively investigated with respect to their nonlinear responses to electric fields[1,2]. These nonlinear responses are influenced by the electronic and phononic energy structures of the investigated material, respectively. Analogous to this, the nonlinear responses induced by magnetic fields are closely related to the energy structure of the spin system and spin precessions.

Antiferromagnetic spin systems have resonance frequencies on the order of terahertz (THz) due to a strong exchange interaction between neighboring spins. Thus, the ultrafast and nonlinear dynamics of magnetization in such systems have attracted considerable attention from the perspective of fundamental physics and applications in magnonics and spintronics[3–13]. However, due to the rather small magneto-optical susceptibility of these materials, it is relatively difficult to track the nonlinear spin responses by using optical pulses[14–16].

[1]Institute for Chemical Research, Kyoto University, Uji, Kyoto 611-0011, Japan. [2]Department of Basic Science, University of Tokyo, Meguro, Tokyo 153-8902, Japan. [3]Department of Physics, Chiba University, Chiba 263-8522, Japan. [4]Department of Physics, Tokyo Institute of Technology, Tokyo 152-8551, Japan. [5]Department of Electronic Science and Engineering, Kyoto University, Kyoto, Kyoto 615-8510, Japan. [6]Department of Physics, Graduate School of Science, Kyoto University, Kyoto, Kyoto 606-8502, Japan. [7]Laboratory for Materials and Structures, Tokyo Institute of Technology, Yokohama, Kanagawa 226-8503, Japan. [8]Department of Energy and Hydrocarbon Chemistry, Graduate School of Engineering, Kyoto University, Kyoto, Kyoto 615-8510, Japan. ✉e-mail: kanemitu@scl.kyoto-u.ac.jp; hirori@scl.kyoto-u.ac.jp

Since their first use a decade ago, THz pulses have been considered a more efficient means of directly exciting spin waves (magnons) in antiferromagnets for spin control[17–25]. One of the distinct signatures of nonlinear spin dynamics is the generation of second-harmonic (SH) or third-harmonic (TH) oscillations. However, because the maximum peak amplitude of a single-cycle THz pulse usually reaches only about 0.1 T and its spectral density at the spin resonance frequency is also limited, the harmonic spin oscillation is limited to the second order in common approaches[26]. Although it has been shown that a THz electric field enhanced by an antenna structure allows us to rotate the magnetization at low temperatures[27], the resulting spin dynamics showed no higher-order harmonic oscillation near the critical temperature of spin reorientation. Furthermore, the spin dynamics of antiferromagnets are described by two sublattices, and therefore both the ferromagnetic vector $M$ and the antiferromagnetic vector $L$ may contribute to the nonlinear responses[8]. However, the observation of a sizable $L$ at room temperature requires strong excitation. Even a THz magnetic field enhanced by a split ring resonator was not able to reveal obvious nonlinear properties besides a redshift in the magnon frequency;[28] in other words, the excitation was still too weak. Such experimental results can be sufficiently described by the dynamics of $M$ without the need of considering $L$. Thus, it has remained unclear how $M$ and $L$ contribute to the generation of higher-order harmonic spin oscillations in the strong excitation regime and nonlinear magnetization changes in general. In this study, we observed a large spin-precession amplitude that generates the SH and TH of spin motion in the canted antiferromagnet $HoFeO_3$ by using a large THz magnetic near field, and clarified their relation with the dynamics of the ferromagnetic vector $M$ but also the antiferromagnetic vector $L$.

## Results

### Magnetic field enhancement in a spiral-shaped microstructure

The measurement geometry is presented in Fig. 1a (Methods and Supplementary Section I). The sample (shown in blue) was a 52-μm-thick $c$-cut $HoFeO_3$ crystal with an intrinsic quasi-antiferromagnetic (q-AF) mode at $\nu_{AF} = 0.58$ THz. The linearly polarized electric field of a THz pulse (red thick arrow) was coupled to the long triangular tail of a metallic spiral-shaped microstructure on the surface of the sample (shown in yellow). The original waveform of the THz pulse is shown by the gray curve in Fig. 1b[29]. To remove the field components that are not needed to excite the q-AF mode, a low-pass filter was inserted in front of the sample (cut-off frequency: 0.68 THz; the black curve in Fig. 1b is the obtained time trace). An image of the fabricated structure is shown in Fig. 1c. The THz pulse-induced current flows into the spiral of the microstructure, which enhances the THz magnetic field. The magnetic field at the center of spiral structure is about 200 times larger than that of the incident THz magnetic field at the resonance frequency $\nu_c$ of the structure [=0.54 THz, which was determined by a finite-difference time-domain (FDTD) simulation]. Compared with the case of using a split ring resonator[28], a three times stronger and spatially smoother magnetic field can be realized by using this structure, as shown at the bottom of Fig. 1c (Supplementary Sections II–IV). Accordingly, we were able to generate magnetic field amplitudes up to 2.1 T even with the filtered THz pulse. The strong magnetic field along the $z$-axis (see the black curve in Fig. 1d) exerts a Zeeman torque on the spins in $HoFeO_3$ and thus induces a change in the net magnetization along the $z$-axis, $\Delta M_z$.

### Asymmetric changes in the Faraday ellipticity

As shown by the thin red arrows in Fig. 1a, a linearly polarized near-infrared (NIR) probe pulse for the Faraday ellipticity measurements is incident from the back of the $HoFeO_3$ crystal and focused at the center of the spiral. The measurement is based on the fact that a change in the magnetization leads to a change in the polarization ellipticity angle of the probe light that passes through the crystal (Supplementary Section I). The three temporal profiles of the change in the ellipticity angle ($\Delta\eta$) in Fig. 1d for THz electric field amplitudes equal to $E_{THz} = 0.8$, 0.55, and 0.2 MV cm$^{-1}$ correspond to the data for $B_z = 2.1$, 1.4, and 0.5 T at the sample surface, respectively. We can see

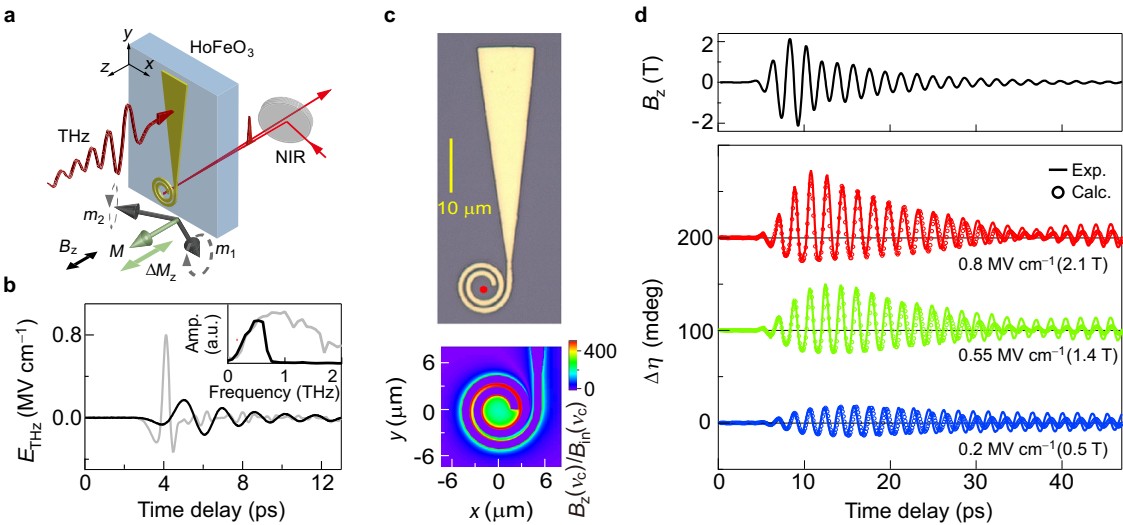

**Fig. 1 | Faraday ellipticity measurements. a** Schematic diagram of the experiment. A probe pulse with a wavelength of 800 nm is focused at the center of the spiral by an objective lens. The gray arrows at the bottom of the figure represent the sublattice magnetizations $m_1$ and $m_2$, and the green arrow is the ferromagnetic vector $M = m_1 + m_2$. **b** The gray and black curves are the time-domain profiles of the THz pulse before and after the low-pass filter. The corresponding Fourier transform spectra are shown in the inset. **c** A microscopic image of the gold microstructure is shown at the top, and the red solid circle indicates the position at which the ellipticity change was probed. The calculated distribution of the enhancement factor of the magnetic field at the spiral resonance frequency

$\nu_c = 0.54$ THz, $B_z(\nu_c)/B_{in}(\nu_c)$, is shown at the bottom, where $B_z$ is the magnetic field enhanced by the microstructure and $B_{in}$ is the incident THz magnetic field strength after the low-pass filter. **d** The black curve in the upper panel is the magnetic field at the center of the spiral at the sample surface calculated by a finite-difference time-domain simulation. The red, green, and blue curves show the observed time-domain signals of the change in the Faraday ellipticity angle $\Delta\eta$ for THz pulses with $E_{THz} = 0.8$, 0.55, and 0.2 MV cm$^{-1}$, respectively. The red and green curves are offset for clarity. The open circles are the corresponding results calculated by using the LLG equation and considering the magneto-optical effect. Note that "a.u." is the abbreviation for "arbitrary units".

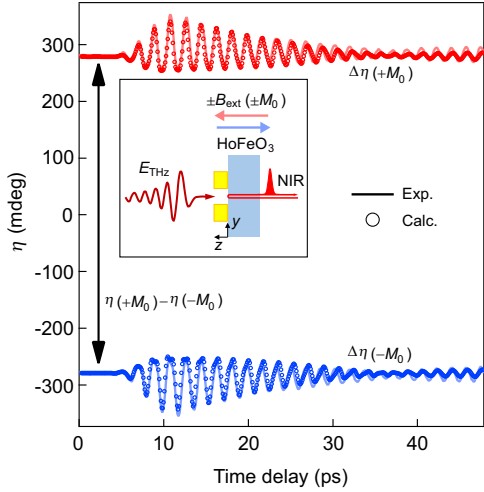

**Fig. 2 | Dependence of the Faraday ellipticity angle on the initial magnetization direction.** The red and blue curves are the Faraday ellipticity angles measured using an initial magnetization of $+M_0$ and $-M_0$, respectively. The circles are the corresponding calculation results. The calculations are consistent with the experimental data, which indicates that not only the dynamic $\Delta\eta$ but also the static value $\eta(+M_0)-\eta(-M_0)$ can be reproduced by our calculation. In the experiments, the initial magnetization ($\pm M_0$) was switched by applying an external static magnetic field ($\pm B_{ext}$).

that the curve for $E_{THz} = 0.8\,MV\,cm^{-1}$ exhibits a distorted sinusoidal oscillation with an asymmetric amplitude distribution; i.e., the amplitudes of the signals with $\Delta\eta > 0$ are larger than those of the signals with $\Delta\eta < 0$.

The asymmetric waveform of $\Delta\eta$ for $0.8\,MV\,cm^{-1}$ observed over the course of about 30 ps is a fingerprint of a large magnetization change induced by the high magnetic field. This data is not a result of the THz Kerr effect, because the Kerr effect is attributed to the diagonal elements of the third-order nonlinear susceptibility $\chi^{(3)}$ and thus should not depend on the sign of the magnetization, but our measured $\Delta\eta$ depends on the sign as shown in Fig. 2 and Supplementary Section V. We can also exclude thermally induced changes in the spin structure, such as demagnetization or spin reorientation, because their recovery times (to reach the equilibrium state) are much longer than tens of picoseconds[30].

## Data analysis using ferromagnetic and antiferromagnetic vectors
To explain the waveform of $\Delta\eta$ and to evaluate the magnetization change achieved by our method, we numerically solved the Landau–Lifshitz–Gilbert (LLG) equation and the propagation equation of the probe pulse (Supplementary Sections VI and VII). The former equation was used to obtain the dynamics of the sublattice magnetizations $m_i$ with $i = 1, 2$, and the latter equation was used to calculate the Faraday ellipticity change occurring in the sample. The propagation equation includes the following magnetization-dependent permittivity tensor for $HoFeO_3$:[31]

$$\varepsilon = \begin{pmatrix} \varepsilon_0 + \sigma & i\kappa \\ -i\kappa & \varepsilon_0 - \sigma \end{pmatrix}. \qquad (1)$$

Here, the parameters $\varepsilon_0$ and $\sigma$ are magnetization-independent terms, whereas the off-diagonal term $\kappa$ is a magnetization-dependent term. $\sigma$ describes the strength of the birefringence. Regarding $\kappa$, in our calculation, the contributions from both the ferromagnetic vector $M = m_1 + m_2$ and the antiferromagnetic vector $L = m_1 - m_2$ are taken into account (note that $L$ can only be ignored in the case of small

deflection angles):[31]

$$\kappa = f\frac{M_0 + \Delta M_z}{M_0} + g\frac{L_0 + \Delta L_x}{L_0}, \qquad (2)$$

where $\Delta M_z$ ($\Delta L_x$) is the temporal variation of the $z$-axis ($x$-axis) component of $M$ ($L$). $M_0$ and $L_0$ are the initial amplitudes of these vectors. $f$ and $g$ represent phenomenological constants. The calculated waveforms for different magnetic field strengths are shown by the open circles in Fig. 1d, which reveals that our model reproduces the experimental data (solid curves) well. Our calculation also reproduces the absolute values of $\eta$ for different signs of the initial magnetization, $\pm M_0$ (Fig. 2).

## Interpretation of the asymmetric temporal profiles
The good agreement between the experiment and the calculation allows us to understand the origin of the asymmetric temporal profiles of $\Delta\eta$: they originate from spin dynamics with a large deflection angle. The calculated dynamics of $\Delta M_z/M_0$ and $\Delta L_x/L_0$ are presented in Fig. 3a, b, respectively. In stark contrast to the behavior of $\Delta M_z/M_0$, $\Delta L_x/L_0$ shows a vertically asymmetrical behavior. This difference can be graphically explained by the motion of $L$: The red (blue) trajectories in Fig. 3c show a condition where the spins are strongly (weakly) excited and deviate from the $x$-axis by a large (small) maximum deflection angle $\theta$. While $M$ oscillates and its amplitude changes, $L$ rotates around the $z$-axis by $\pm\theta$ (with an almost constant $|L|$ due to the small canting angles of $m_i$; see Supplementary Section VIII). The rotation of $L$ is shown in Fig. 3d (the top view of Fig. 3c). As shown by the calculated results in Fig. 3b, the $x$ component of $L$ always decreases as the deflection angle increases and $\Delta L_x$ oscillates with twice the frequency of $\Delta M_z$. The difference in the oscillation frequency is explained in Fig. 3d: the change in the $x$ component of $L$ always proceeds along the same path ($x = 2 \to 2-|\Delta L_x| \to 2$), independently of the path of the arrowhead of the vector $L$ projected on the $y$-axis (either $y = 0 \to +|\Delta L_y| \to 0$ or $0 \to -|\Delta L_y| \to 0$).

The field-strength dependences of their maximum values ($\Delta M_z^{max}/M_0$, $\Delta L_x^{max}/L_0$, and $\theta$) are presented in Fig. 3e, where $\theta$ almost linearly depends on $E_{THz}$ and $\Delta M_z^{max}/M_0$ increases approximately linearly with $\theta$. However, because $\Delta L_x^{max}/L_0$ exhibits a cosine relationship with $\theta$, it is much less sensitive to changes in $\theta$ as long as $\theta$ is small. Thus, only if the spin precession is driven far away from the equilibrium state, a considerable change in $\Delta L_x/L_0$ appears and contributes to $\Delta\eta$, causing the asymmetry shown in Fig. 1d. The data points indicated by the black arrows in Fig. 3e represent the values calculated for the electric fields used in the experiment. From the results, we determine $\theta \approx 40°$, $\Delta M_z^{max}/M_0 \approx 0.9$ and $\Delta L_x^{max}/L_0 \approx 0.2$ at the surface of $HoFeO_3$ for $E_{THz} = 0.8\,MV\,cm^{-1}$. In the experiment in ref. [28], the change in the $x$ component of $L$ ($\Delta L_x$) reached only $\approx 3.8\%$ even though the change in the $z$ component of $M$ ($\Delta M_z$) reached $\approx 40\%$. These experimental results can be sufficiently accurately reproduced by the dynamics of $M$ without the need of considering $L$, because $L$ (Fig. 3e; green curve) increases more slowly than $M$ (Fig. 3e; blue curve) in the weak excitation regime.

## Higher-order harmonic spectra in the time-frequency domain
In addition to the observed asymmetric change in magnetization, the generation of an SH or TH spin oscillation is a hallmark of a large magnetization amplitude far away from the equilibrium value $M_0$. As shown in Fig. 4a, the Fourier transform spectra of the experimentally obtained $\Delta\eta(t)$ reveal the nonlinearity of the spin motion caused by the enhanced THz magnetic field. In the spectrum for $E_{THz} = 0.2\,MV\,cm^{-1}$ (blue spectrum), there is a fundamental peak at $\nu = 0.58\,THz$, which is equal to the q-AF mode frequency $\nu_{AF}$, and another peak lies at 1.16 THz, which precisely matches the SH of $\nu_{AF}$. In the spectrum for $E_{THz} = 0.8\,MV\,cm^{-1}$ (red shaded area in Fig. 4a), there is a peak

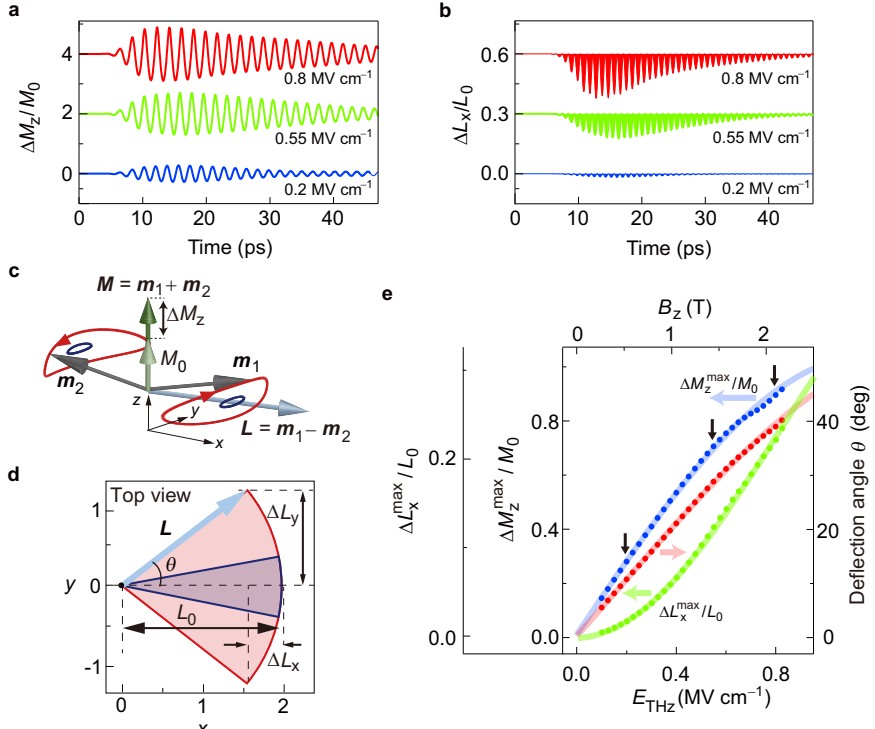

**Fig. 3 | Analysis of magnetization changes $\Delta M_z/M_0$ and $\Delta L_x/L_0$ based on the LLG equation. a, b** Calculated dynamics of $\Delta M_z/M_0$ (**a**) and $\Delta L_x/L_0$ (**b**) for $E_{THz}$ = 0.8, 0.55, and 0.2 MV cm$^{-1}$. The red and green curves are offset for clarity. **c** The definitions of the ferromagnetic vector $\boldsymbol{M}$ (green arrow) and the antiferromagnetic vector $\boldsymbol{L}$ (blue arrow) are shown. The red and blue curves describe the precession motion of the sublattice magnetization ($\boldsymbol{m}_1$ and $\boldsymbol{m}_2$; gray arrows) for $E_{THz}$ = 0.8 and 0.2 MV cm$^{-1}$, respectively. Because the $z$-axis component of the actual trajectory is much smaller than the components in the $x$–$y$ plane ($|M_z| \ll |\triangle L_x|, |\triangle L_y|$), the trajectories are magnified along the $z$-axis for clarity. **d** Top view of (**c**). The gray arrows

are the projections of $\boldsymbol{m}_1$ and $\boldsymbol{m}_2$ on the $x$–$y$ plane. The light red and blue shaded sectors indicate the angular range of motion for $E_{THz}$ = 0.8 and 0.2 MV cm$^{-1}$, respectively. The deflection angle $\theta$ is defined as half of the central angle of a sector. **e** Calculated $E_{THz}$ dependence of $\theta$ (red points), $\Delta M_z^{max}/M_0$ (blue points), and $\Delta L_x^{max}/L_0$ (green points). The black arrows indicate the results for $E_{THz}$ = 0.8, 0.55, and 0.2 MV cm$^{-1}$, respectively. The solid curves are guides to the eye. The results in (**a**, **b**, **e**) are the numerical solutions of the LLG equation at the HoFeO$_3$ surface, including the microstructure.

corresponding to the TH, in addition to the fundamental and SH peaks. However, compared with the spectrum measured at weaker fields (blue shaded area in Fig. 4a), the center frequencies of the SH (1.06 THz) and TH (1.58 THz) peaks are slightly lower than $2\nu_{AF}$ (1.16 THz) and $3\nu_{AF}$ (1.74 THz), and the three peaks exhibit an obvious broadening towards the low-frequency side.

To elaborate on the observed behavior of the higher-order harmonic oscillations, we examine the time-resolved Fourier transform spectra of $\Delta\eta(t)$ for $E_{THz}$ = 0.8 and 0.2 MV cm$^{-1}$ in Fig. 4b. While the maximum amplitude of the fundamental peak becomes stronger as $E_{THz}$ increases, the rise time and decay time become shorter. To focus on the signal duration, the temporal dynamics of the normalized fundamental, SH, and TH signals obtained by integrating the peak area are shown in Fig. 4c.(the data without normalization are shown in Supplementary Section IX). The signals of the SH and TH peaks reach their maxima at almost the same time as the fundamental peak, but they decay faster, verifying that the SH and TH are indeed generated only when the fundamental magnetization change is large. The spectral position of the fundamental peak as a function of time is shown in Fig. 4d.at early times before 20 ps, the oscillation peak frequencies notably deviate from $\nu_{AF}$ and the oscillation peak frequency even reaches $\nu_{red}$ = 0.53 THz for $E_{THz}$ = 0.8 MV cm$^{-1}$. This redshift is a result of the intrinsic nonlinear properties of the antiferromagnet in addition to the effect of the forced oscillation of the spin in the THz magnetic field at $\nu_c$ = 0.54 THz (Supplementary Sections X and XI). As shown in Fig. 4a, because of the redshift to the frequency $\nu_{red}$ (0.53 THz), the SH and TH peaks are almost centered at $2\nu_{red}$ (1.06 THz) and $3\nu_{red}$ (1.58 THz), respectively.

## Mechanisms responsible for the decay of the observed Faraday signals

The faster decay before 25 ps (Fig. 4c) of the fundamental oscillation amplitude at higher excitation intensities is attributed to two mechanisms. One mechanism is the magnetization-dependent Gilbert damping (Supplementary Section VI), which represents the decay rate of spin precession to the equilibrium state proportional to the magnetization change. Owing to this damping, a larger magnetization change can lead to faster decay. The second mechanism is considered to be the interference between oscillation components with different frequencies. When the probe pulses are transmitted through the sample, they experience different degrees of magnetization change because of the inhomogeneous distribution of the magnetic field along the $z$-axis (Fig. S4 in Supplementary Section II). Additionally, we need to consider that the frequency of spin precession shifts depending on the magnitude of the magnetization change. The interference among the components with different frequencies leads to a beating wave (Supplementary Sections XII and XIII) and thus results in a faster decay of the Faraday signal at higher excitation intensities.

Furthermore, the observation of an $L$-change-induced asymmetric oscillation in $\Delta\eta$ motivates an investigation of the contributions of both $M$ and $L$ to the harmonic oscillations. Figure 4e shows the spectra of theoretically predicted $\Delta\eta(t)$ results obtained under different assumptions. The blue and red curves are the spectra of the calculated results in Fig. 1d that reproduce the data for $E_{THz}$ = 0.2 and 0.8 MV cm$^{-1}$, respectively. On the other hand, the black dashed curve is the spectrum for $E_{THz}$ = 0.8 MV cm$^{-1}$ obtained by fixing $\Delta L_x/L_0$ to zero.

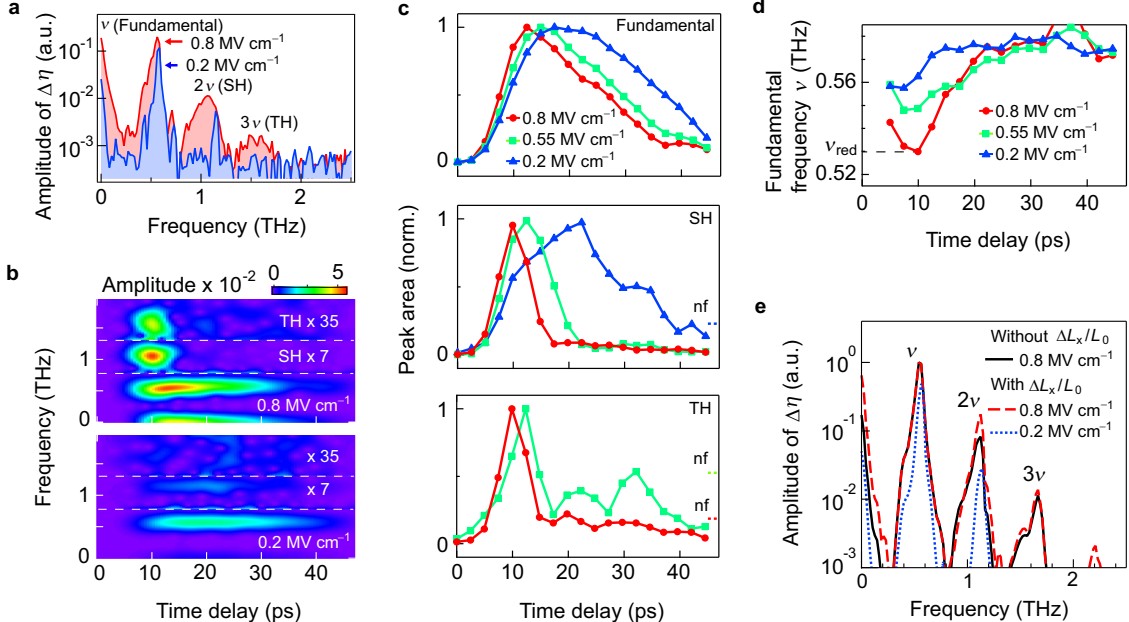

**Fig. 4 | Spectra of the experimentally and theoretically obtained Faraday ellipticity angle change $\Delta\eta(t)$ results. a** The red and blue spectra are the Fourier spectra of the experimental data obtained using $E_{THz}$ = 0.8 and 0.2 MV cm$^{-1}$, respectively. **b** Color maps of the time-resolved Fourier transform spectra of $\Delta\eta(t)$ for $E_{THz}$ = 0.8 MV cm$^{-1}$ (upper panel) and 0.2 MV cm$^{-1}$ (lower panel). The time window used for this calculation is a Gaussian function with a width of 5 ps and all color maps use the same color scale. **c** Normalized temporal variations of the fundamental (top), SH (middle), and TH signals (bottom), where the data points (circle: 0.8 MV cm$^{-1}$, square: 0.55 MV cm$^{-1}$, triangle: 0.2 MV cm$^{-1}$) are the peak areas of the corresponding peaks in each time slice in (**b**). The noise floor (nf) levels are the

maximum integration values in the corresponding spectral range that were observed after the peaks had vanished. **d** Temporal variation of the oscillation frequency obtained from the spectral position of the fundamental peak in each time slice for the data (circle: 0.8 MV cm$^{-1}$, square: 0.55 MV cm$^{-1}$, triangle: 0.2 MV cm$^{-1}$) in (**b**). **e** Spectra of theoretically predicted $\Delta\eta(t)$ results under different assumptions. The black curve is the spectrum for $E_{THz}$ = 0.8 MV cm$^{-1}$ obtained in the case that $\Delta L_x/L_0$ is fixed at a value of zero, while the red dashed and blue dotted curves are, respectively, the spectra for $E_{THz}$ = 0.8 and 0.2 MV cm$^{-1}$ obtained without fixing $\Delta L_x/L_0$ to zero, respectively. Note that "a.u." is the abbreviation for "arbitrary units".

The comparison between the red and black dashed curves indicates that the observed SH peak at $2\nu$ is caused not only by the second-order harmonic oscillation of $\Delta M_z/M_0$, but also by the fundamental oscillation of $\Delta L_x/L_0$ (Supplementary Section VIII). This is because $\Delta L_x/L_0$ oscillates at twice the frequency of $\Delta M_z/M_0$ and thus contributes to the even-order harmonic oscillations. Meanwhile, the odd higher-order harmonics originate only from the nonlinear response of $\Delta M_z/M_0$. Thus, it can be considered that the observation of the third harmonic is important: it helps us to differentiate between the nonlinearities of the **M** and **L** contributions, because the third harmonic peak is mainly determined by the oscillation of $M_z$ (as shown in Fig. S10 of Supplementary Section VIII). The second and fourth harmonic peaks contain significant contributions from both $M_z$ and $L_x$. Thus, to obtain clearer experimental evidence that shows the nonlinearity of spin precession, the observation of the third harmonic spin oscillation is important.

**Relation between the fundamental, SH and TH amplitudes**

In HoFeO$_3$, the Dzyaloshinskii–Moriya interaction leads to a canting of the sublattice magnetizations. Thus, the potential becomes anharmonic and the system has a broken symmetry, which allows even- and odd-order harmonic oscillations to be generated. To confirm that the observed harmonics in Fig. 4 originate from the nonlinearity of the spin response (and not from other effects such as a nonlinear response of the metallic structure), let us examine the dependence of the SH and TH signals on the fundamental oscillation. As shown in Fig. S16 (Supplementary Section XIV), the faster decay verified in Fig. 4c causes a deviation of the field-strength dependence of harmonics from the power law of the electric (magnetic) field. Each peak-area value plotted in Fig. S16 was obtained from the Fourier transform of the $\Delta\eta$ waveform extending over 47 ps, and thus reflects both the amplitude and

the lifetime of the fundamental, SH and TH components. As the excitation becomes stronger, the resulting faster decay of each component suppresses the increase in the corresponding peak area more effectively. This provides an intuitive explanation of the observed field-strength dependence. On the other hand, the SH and TH amplitudes, that is, the harmonics with order $n$ = 2 and 3, closely follow the $n$th power of the fundamental peak, as shown in Fig. 5. The observed dependences are well reproduced by our calculations. These results indicate that, while the strong excitation condition modifies the field-strength dependence of the harmonics, the observed harmonics truly originate from a large magnetization change. In addition, our observation of the SH and TH is consistent with the selection rule derived from microscopic considerations on the canted antiferromagnet (Supplementary Sections XV and XVI).

In conclusion, owing to the strong excitation, harmonic oscillations up to the third order were observed. In particular, by realizing large magnetization changes, we were able to observe an **L**-change-induced asymmetric oscillation. We have explained that the fundamental oscillation of $\Delta L_x/L_0$ contributes to the observed SH peak in addition to the second-order harmonic oscillation of $\Delta M_z/M_0$, but it does not contribute to the TH because $\Delta L_x/L_0$ oscillates at $2\nu$. In addition, the ability to induce a large change in **L** is considered very important for future applications of ultrafast spintronics, because the absolute value of **L** is much larger than that of **M**, and thus the dynamics of **L** determine the spin current injected by spin precession[10,11,32]. Beyond providing an understanding of the spin nonlinearity and potential applications of antiferromagnets, our achievements indicate that our method may serve as a tool to control the functional properties of solids, including ferrimagnets, multiferroelectric materials, and quantum spin systems[18,33,34], by strong THz magnetic near fields.

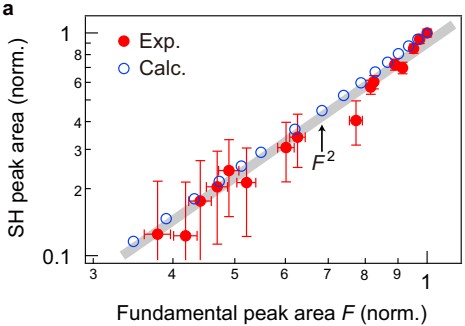

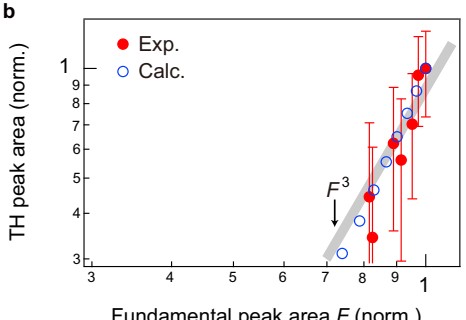

**Fig. 5 | Dependence of the higher harmonic peak area on the fundamental peak area. a, b** Normalized dependence of the SH (**a**) and TH peak area (**b**) on the fundamental peak area $F$. The data points for the experiment (solid circles) and calculation (open circles) are the peak areas obtained by integrating the corresponding

Fourier transform amplitudes in the spectra. The error bars represent the noise floor in the spectra, which is obtained by integrating the Fourier transform amplitudes of the case for $E_{THz} = 0.2$ MV cm$^{-1}$ over the range from 1.25 to 1.71 THz. The gray lines are proportional to the square and cube of the fundamental peak area $F$.

## Methods

### Sample properties and spin dynamics in a HoFeO$_3$ crystal

We used a HoFeO$_3$ single crystal with a $c$-cut surface in the *Pbnm* setting[35]. The crystallographic $a$-, $b$-, and $c$-axes are parallel to the $x$-, $y$-, and $z$-axes, respectively. The HoFeO$_3$ single crystal was grown by using the floating-zone method and polished to a thickness of 52 µm. A single magnetic domain was confirmed by a two-dimensional static Kerr ellipticity measurement at zero magnetic field. Before each experiment, we applied a DC magnetic field to the sample to saturate its magnetization along the $z$-axis. The hysteresis in the Kerr rotation plotted versus the magnetic field is shown in Supplementary Section I. We fabricated an array of gold microstructures with a thickness of 200 nm on the crystal surface by an electron-beam lithographic process.

At room temperature, the two magnetizations $\boldsymbol{m}_i$ ($i$ = 1, 2) of the iron sublattices in HoFeO$_3$ are mainly antiferromagnetically aligned along the $x$-axis (with a slight canting angle $\beta_0 = 0.63°$ due to the Dzyaloshinskii–Moriya field) and exhibit a spontaneous magnetization $\boldsymbol{M}$ along the $z$-axis (see the two gray arrows and the green arrow below the sample in Fig. 1a)[36]. In the THz region, there are two antiferromagnetic resonance modes, the so-called quasi-antiferromagnetic (q-AF) mode and the quasi-ferromagnetic (q-F) mode. The magnetic near field $B_z$ in our setup causes a q-AF-mode motion; as illustrated by the gray dashed curves in Fig. 1a, the Zeeman torque exerted on the spins triggers a precessional motion of each sublattice magnetization $\boldsymbol{m}_i$ about the equilibrium direction. This precession causes an oscillation of the macroscopic magnetization $\boldsymbol{M} = \boldsymbol{m}_1 + \boldsymbol{m}_2$ in the $z$-direction, as shown by the green double-headed arrow. The resultant magnetization change in the $z$-direction, $\Delta M_z$, is detected as the Faraday ellipticity signal $\Delta\eta$.

### Faraday ellipticity measurements

In this experiment, we used a THz electric field to induce an oscillating current in the spiral-shaped structure and generate a strong magnetic near field $B_z$ perpendicular to the HoFeO$_3$ crystal surface. The magnetization change induced by the THz magnetic field is recorded by time-resolved measurements of the Faraday ellipticity. An amplified Ti:sapphire laser (repetition rate 1 kHz, central wavelength 800 nm, pulse duration 80 fs, and 7 mJ/pulse) was used to generate intense THz pulses by optical rectification of the femtosecond laser pulses in a LiNbO$_3$ crystal used in the tilted-pump-pulse-front scheme. The THz pulses were focused on the gold microstructure by using an off-axis parabolic mirror with a focal length of 50 mm, resulting in a spot diameter of ≈300 µm (full width at half-maximum). As shown in the inset of Fig. 1b, the spectrum of the incident-filtered THz electric field had a peak at around 0.57 THz. For the time-resolved Faraday ellipticity measurements, we used a 50× objective lens to focus linearly polarized

optical probe pulses in the plane of the microstructure (spot diameter ~1.2 µm), which enabled us to measure local changes in the Faraday ellipticity angle $\eta$. Note that the HoFeO$_3$ crystal was sufficiently transparent for the used 800-nm probe pulses. The polarization directions of the THz light and the probe light were parallel and almost along the $y$-axis. A balanced detector combined with a quarter-wave plate and a Wollaston prism was used to measure the ellipticity angle of the reflected probe pulse. The detection geometry and the definition of $\eta$ are shown in Fig. S1(Supplementary Section I). All experiments in this study were performed at room temperature.

## Data availability

Source data are available for this paper. All other data that support the plots within this paper and other findings of this study are available from the corresponding author upon request. Source data are provided with this paper.

## Code availability

The code used to simulate the magnetization dynamics and the Faraday ellipticity change is available from the corresponding author upon request.

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

## Acknowledgements
The authors thank Motoaki Bamba for fruitful discussions. Part of this study was supported by a grant from the Japan Society for the Promotion of Science, JSPS KAKENHI Grants JP19H05465 (Y.K.), JP19H05824 (H.H.), and JP21H01842 (H.H.). This work was also supported by JST SPRING Grant JPMJSP2110 (Z.Z.).

## Author contributions
Z.Z. and H.H. carried out the experiments. Z.Z., F.S., T.M., S.C.F., M.S., T.S., Y.K., and H.H. analysed the data. S.C.F. and M.S. developed the theory of the selection rules for the high harmonic generation of spin dynamics. Z.Z., Y.K., and H.H. designed the metallic microstructure. Y.M., K.T., T.Y., H.K., and H.H. fabricated the HoFeO$_3$ crystal. Y.K. and H.H. conceived and supervised the project. All authors discussed the results and contributed to the writing of the paper.

## Competing interests
The authors declare no competing interests.
