## [Peer Review File · Nature Communications]

Generation of third-harmonic spin oscillation from strong spin precession induced by terahertz magnetic near fieldsTransfer to *Nature Communications* (Manuscript NPHOT-2022-06-00849A-Z)

Title: Generation of third-harmonic spin oscillation from strong spin precession induced by terahertz magnetic near fields

Author: Z.-Y. Zhang et al.

General response to the recommendations and criticisms:

We would like to thank all reviewers for carefully reading our revised manuscript. The comments provided by the reviewers were very helpful to improve our work further, as explained in the point-by-point responses below. To draw the focus on the technically important features for specialists, we decided to transfer our manuscript to a more specialized Journal (*Nature Communications*). One technically important feature is the observation of third-order harmonic spin oscillations in antiferromagnets induced by large magnetization changes, where both the ferromagnetic and antiferromagnetic vectors can contribute significantly to the harmonics. Beyond providing an understanding of the spin nonlinearity in antiferromagnets for future spintronics and magnonics applications, this manuscript shows a way to control the functional properties of solids by THz magnetic fields.

The major revisions explained in the following are highlighted in blue in the revised manuscript for your convenience. Also we added and modified new Figures according to the comments. We would be grateful if you decide to review our manuscript for publication in *Nature Communications*.

Figure updates

<Main text>

Figure 3: We modified the size of graphs without any changes of content.

Figure 4: We modified the size of graphs without any changes of content.

<Supplementary Information>

Figure S10: We added a graph regarding the Fourier spectra of the $\Delta M(t)$ and $\Delta L(t)$.

Figure S11: We moved a graph regarding the temporal variations of the fundamental, SH and TH without normalization from the main text.

Figure S12: We moved a graph regarding the time-integrated spectra of the magnetic field and the q-AF modes from the main text.

Figure S15: We added a graph regarding the origin of the beating signal.

Figure S16: We moved a graph regarding the field-strength dependence for the fundamental, SH and TH from the main text.

Figure S18: We added a graph regarding the symmetry breaking of the magnetic potential.

Replies to Reviewer 1's comments

Comment 1-1

The authors address my previous comments and questions well. I also found the other three referees raised a bundle of critical questions on the work, especially on the novelty. The new version has improved greatly on the novelty expression. I think it is much better. There are some tiny issues that should be addressed further.

Response 1-1

We thank Reviewer 1 for acknowledging the novelty of our work. We hope that the results can be published soon in *Nature Communications*.

Comment 1-2

1) The response in comment 1-7 just explained that the recovery time is longer than 1.2 ns for HoFeO₃, the authors need to distinguish this behavior between ferromagnets (ferrimagnets) and antiferromagnets.

Response 1-2

We thank Reviewer 1 for this comment related to his/her comment in the first review round: "... I think the statement is appropriate for ferromagnet, but I wonder whether it is also true for AFM? Even for ferrimagnets, such as YIG, the heating effect is also very fast. [Nat. Commun. 9, 2899 (2018)]". In the above comment 1-2, Reviewer 1 seems to agree that our statement is true, but additionally asks for pointing out the different behaviour in ferrimagnets. We added the reference Nat. Commun. 9, 2899 (2018) to the conclusion of the manuscript as Ref. 33, where we clarify that the application of our method to other materials such as ferrimagnets should be considered in the future. On the other hand, we did not change the explanation of Fig. 1 due to the following reasons:

In Nat. Commun. 9, 2899 (2018), Seifert et al. mentioned the typical time for energy transfer from the electron system to the phonon system. On the other hand, the observed asymmetric temporal behavior in Fig. 1d does not originate from sample heating. Because this temporal behavior is not determined by the electron-phonon interaction described by Seifert et al., we think that comparing our result with that reported by Seifert et al. may be confusing for the audience. Additionally, the sample used by Seifert et al. is not a single crystal, but consists of YIG (3 μm) and Pt (5.5 nm) films. This difference in the sample structure may modify the thermal recovery time, but this is not discussed in Nat. Commun. 9, 2899 (2018).

Comment 1-3

2) The static magnetic field of the Kerr rotation of HoFeO₃ needs to be marked.

I recommend its publication after properly considering the remaining points above.

Response 1-3

We agree and thus added the following statement to the Methods section: *The hysteresis in the Kerr rotation plotted versus the magnetic field is shown in the Supplementary Section I.*

Replies to Reviewer 2's comments

Comment 2-1

I have read the revision carefully and I appreciate the authors take the time to answer my questions/comments. However, my assessment is consistent with the prior reports. The overall data quality and experimental scheme are satisfactory but there is lacking enough strikingly new observations and insights. The idea of magnetic near-field excitations has been demonstrated in Ref. 28. The only way to make the current paper to a journal like Nature Photonics to have some observations of much higher nonlinearities that have not achieved in any other means. However, although the observation of 3rd harmonic generation is new, the nonlinear mechanism is still on the same order of nonlinearity as the pump-probe and two-quanta processes discussed in Ref. 19. It could be because the M and L enhancement is still on the same order of prior work, e.g., they change about 2 and 6-7 times higher. These are nice technical improvements but I still believe the excitement and impact are relatively low and the paper is more towards incremental technical progress that is hard to justify itself the need to be broadly digested in the community like those papers normally seen in

Nature Photonics. I'm sorry to say that I cannot recommend for publication.

Response 2-1

We thank Reviewer 2 for acknowledging that the overall data quality and experimental scheme are satisfactory. As explained in detail below, we improved the manuscript according to your suggestions and hope that the revised manuscript is acceptable for publication in *Nature Communications*.

Comment 2-2

Some other questions relate to M and L interaction. The experiment is done at room temperature above T_c the results should be mostly determined by the L and I don't understand how the M and L coupling can contribute significantly. Some clarifications may be needed for the sake of community.

Response 2-2

We thank Reviewer 2 for this question, which points out that the details of coupling unclear. We added a reference before Eq. (2) to solve this issue. Below, we briefly explain why the magnetization dynamics should be described by M and L .

Firstly, we would like to clarify that, due to the constraints $M^2+L^2=const.$ and $M\cdot L=0$, the magnetization dynamics can be described by either M or L as shown in the Supplementary Information of Ref. 14 (present Ref. 22) [Nat. Mater. 20, 607 (2021)]. In other words, M and L are equivalent for the description of the magnetization dynamics, and microscopic interactions and coupling between them do not exist except for the above-mentioned constraints.

Secondly, in our experiment, the magnetization dynamics are observed using the magneto-optic effect (i.e., we measured $\Delta\eta$). The probe-pulse polarization changes according to the degree of magnetization, and the permittivity tensor in the propagation equation of the probe pulse depends on M and L , which is shown in Eqs. (1) and (2) in the main text and also in Ref. 31 [Phys. Rev. B. 84, 064402 (2011), Table I, AFmode, ϵ_{xv}^a]. Actually, as shown in Fig. 3b, L has asymmetric temporal shape and oscillates with a $2v$ fundamental frequency, but as shown in Fig. 1d the experimental Faraday signals contain the symmetric component and oscillations with a v fundamental frequency. Thus only L can not reproduce the experimental results and both M and L contribute to the results. We added Ref. 31 just before Eq. (2) in the main text to show where the classification of the relation between the permittivity tensor and the components of M and L can be found.

Comment 2-3

I'm also not so sure why authors consider the Ref. 13 is about the electric field effect not magnetic field. A large spin rotation is reported in Ref. 13 using intense THz pumping but large B field is together with the large E field. If one uses numbers in Ref 13 one can already get magnetic field is one the same Tesla level achieved in this paper. Therefore, I feel the authors may oversell their work by not giving enough credit to the literature.

Response 2-3

We consider that Ref. 13 (present Ref. 16) [Schlauderer et al., Nature 569, 383 (2019)] shows the effect of the electric field, because Schlauderer et al. explicitly stated that the magnetic field is not enhanced. The magnetic field thus has an amplitude of less than 0.4 T in the case of an incident electric field of 1.2 MV/cm, which is much smaller than the 2.1 T in our work. To explain the details, we briefly review the contents of Ref. 13:

Schlauderer et al. used the electric field enhanced in the gap of a custom-made antenna structure to induce a large spin rotation. On page 2 in Ref. 13, it is written: "*Here we combine the advantages of electric-field-induced anisotropy changes in an antiferromagnet with the local near-field enhancement of metal antennas.*", and on page 3, Schlauderer et al. wrote: "*whereas the q-afm magnon can only be launched by Zeeman coupling to the THz magnetic field, which is not enhanced in the feed gap.*" This is a logical consequence of the used antenna structure, because an enhancement of the magnetic field requires a current flow (as in a split ring resonator or the spiral structure in our present work), and an antenna structure cannot allow a current flow and an magnetic-field enhancement in the gap. Therefore, we believe that our explanations are consistent with the literature.

Comment 2-4

Some other technical issues still exist in the manuscript. For example, the physical origin of the decay

process was very speculative and I don't see why the strong excitation will decay faster than the weak perturbation.

Response 2-4

We thank Reviewer 2 for clarifying his/her concerns regarding the explanation of our new experimental data. It is true that the explanation based on the magnetization-dependent Gilbert damping is speculative and still under discussion. On the other hand, it is a matter of fact that such a mechanism or a similar mechanism is needed to explain the experimental data sufficiently well:

In Ref. 28, the magnetization dynamics of HoFeO₃ at the sample surface were measured. For clarification, we show Fig. 3d of Ref. 28 in Fig. R1 below. The black broken curve is for the case of a constant Gilbert damping, and it cannot reproduce the red data well. As shown by the red solid curve, a magnetization-dependent Gilbert damping is able to reproduce the data well, which is important information for further investigations on the physical origin.

Fig. R1. Time evolution of the oscillation amplitude of the detected Faraday rotation. The red and blue points are the experimental data for high and low pump fluences, respectively. The red and blue curves are the corresponding theoretical results obtained by considering a magnetization-dependent damping.

Comment 2-5

In addition, Fig. R5 is confusing to me. It seems to be the most interesting observation would be the 4th harmonic which differential the M and L contribution. However, the paper does not observe that peak.

Response 2-5

We also consider that the observation of the 4th harmonic oscillation would be interesting and would have helped to improve the impact of this work. However, the signal-to-noise ratio is still not good enough. Nevertheless, we consider that the observation of the 3rd harmonic is also very important: it helps us to differentiate between the nonlinearities of the M and L contributions, because the 3rd harmonic peak is mainly determined by the oscillation of M_z (as shown in Fig. R2). The 2nd and 4th harmonic peaks contain significant contributions from both M_z and L_x . Thus, to obtain a clearer experimental evidence that shows the nonlinearity of spin precession, the observation of the third harmonic spin oscillation is important.

Fig. R2. Theoretical frequency-domain spectra of the oscillations of $\Delta M_z/M_0$ and $\Delta L_x/L_0$. In the case of the 3rd harmonic, the peak amplitude of $\Delta L_x/L_0$ is only 7.5% of that of $\Delta M_z/M_0$. The data in this graph is normalised to the fundamental peak amplitude of $\Delta M_z/M_0$ at frequency ν . The noise floor of our experiment is presented by the grey line.

Comment 2-6

Comment 2-4 on the oscillatory magnetic field effects of rotation spins. I may not make myself clear. I'm not talking about precession but rather the M_z component. The Zeeman torque will have back and forth motion unlike the static torque and how that can induce finite a M_z component essentially at “zero frequency”.

In summary I regret to say that the paper is below the standard of Nature Photonics in my opinion.

Response 2-6

We thank Reviewer 2 for clarifying that he/she had the projection of the rotating spin onto the z-axis in Fig. 3c in mind, and not the precession itself. In the above comment, Reviewer 2 explains the existence of a more or less static (non-zero) M_z component and asks how this can occur although the field is oscillating. We believe that “ M_z component” refers to the $\Delta M_z(t)$ component in our work because M_0 is a consequence of the canting of spins as explained in the manuscript. As shown in Figs. 3a and 3b, the results of $\Delta\eta$ are determined by the dynamics of both \mathbf{M} and \mathbf{L} . The extremely asymmetric oscillation of L_x shown in Fig. 3b leads to the asymmetric component of $\Delta\eta$ shown in Fig. 1d. On the other hand, the oscillation of $\Delta M_z(t)$ (Fig. 3a) is almost symmetric: there is only a slight distortion owing to the anharmonic potential along the z-axis, but this slight distortion hardly contribute to the asymmetric component of $\Delta\eta$ in the temporal profile. This means that the finite M_z component at “zero frequency” occurs due to the anharmonicity.

Replies to Reviewer 3's comments

Comment 3-1

Authors have revised the manuscript significantly which becomes more improved than the original one. In particular, authors tried to highlight the novelty of the present work compared to other recent publications reporting the large spin deflection and/or the nonlinear spin dynamics. Nevertheless, I am still not sure of its novelty.

Response 3-1

We thank Reviewer 3 for acknowledging that the manuscript has improved. Below, we address the remaining concerns of Reviewer 3. We believe that our results constitute an important technical advance, and thus we hope that the revised manuscript can be published in Nature Communications.

Comment 3-2

Figure 4e shows that the third harmonic oscillation turns out to originate from the nonlinear motion of \mathbf{M} not from the antiferromagnetic vector \mathbf{L} .

This puts an emphasis on a magnitude of the modulation of the ferromagnetic vector, and the experimental detection of its third harmonic oscillation, highlighted in the title, becomes simply an issue of an improvement of the signal-to-noise ratio.

Response 3-2

We thank Reviewer 3 for this comment. It seems that one of the remaining concerns of Reviewer 3

Fig. R3. Theoretical frequency-domain spectra of the oscillations of $\Delta M_z/M_0$ and $\Delta L_x/L_0$. In the case of the 3rd harmonic, the peak amplitude of $\Delta L_x/L_0$ is only 7.5% of that of $\Delta M_z/M_0$. The data in this graph is normalised to the fundamental peak amplitude of $\Delta M_z/M_0$ at frequency ν . The noise floor of our experiment is presented by the grey line.

is that our observation is a simple consequence of a relatively good signal-to-noise ratio. While this is true, we believe that this does not diminish the importance of the first observation of the 3rd harmonic. Below, we explain why we consider that the ability to detect the 3rd harmonic has a scientific value:

The observation of the 3rd harmonic is very important, because it helps us to differentiate between the nonlinearities of the \mathbf{M} and \mathbf{L} contributions. This is because the 3rd harmonic is mainly determined by the oscillation of M_z (as shown in Fig. R3), while the 2nd and 4th harmonic peaks contain significant contributions from both M_z and L_x . Thus, to obtain a clearer experimental evidence that shows the nonlinearity of spin precession, the observation of the third harmonic spin oscillation is important.

Comment 3-3

As the third-harmonic oscillation originates from the nonlinear motion of \mathbf{M} not from the antiferromagnetic vector \mathbf{L} , it would be interesting to see that the Fourier-transformed spectra of temporal oscillations in Fig. 2a and b give nonlinear spectral contributions differently. I am not sure whether the theoretical prediction would mean this process or not.

Response 3-3

We thank Reviewer 3 for providing details on his/her considerations regarding the different contributions of \mathbf{M} and \mathbf{L} . The frequency-domain spectra of $\Delta M_z/M_0$ and $\Delta L_x/L_0$ for $E_{\text{THz}} = 0.8$ MV/cm are shown in Fig. R3 presented in the above response. Regarding the contributions of \mathbf{M} and \mathbf{L} to a single peak, we can identify two types of peaks: (1) For the 2ν and 4ν peaks, $\Delta M_z/M_0$ and $\Delta L_x/L_0$ contribute similarly. (2) For the ν and 3ν peaks, the relative contribution of $\Delta L_x/L_0$ is very small and its absolute magnitude lies close to or below the noise floor of our experiment.

The contribution of $\Delta L_x/L_0$ to the fundamental peak can be understood by considering the motion of \mathbf{m}_2 : An exaggerated and strongly magnified image of the motion of \mathbf{m}_2 is shown in Fig. R4, and the projection of \mathbf{m}_2 onto the x - y plane represents the sweep motion of $\mathbf{L}/2$. The frequency of this cycle of \mathbf{m}_2 is ν . During each cycle of \mathbf{m}_2 , $\mathbf{L}/2$ sweeps two times across the x - y plane (from $+y$ to $-y$ and from $-y$ to $+y$), resulting in a frequency of 2ν with respect to the change ΔL_x . However, the two sweep trajectories of $\mathbf{L}/2$ do not completely overlap: the difference is shown by the shaded region in the x - y plane in Fig. R4. Because the difference $\Delta X/2$ only occurs one time for each cycle of \mathbf{m}_2 , the ΔL_x oscillation at 2ν is slightly modulated by an oscillation at ν .

Fig. R4. Schematic of the motions of \mathbf{m}_2 and \mathbf{L} .

Comment 3-4

In line 224, authors should specify which symmetry is broken explicitly.

Considering progresses and sound discussions made in this work, although I would not easily agree its novelty worth to be published in Nature Photonics, I would rather recommend it to be shared in other journals, such as Nature Communications.

Response 3-4

According to this advice, we added the following figure to the Supplementary Information: Figure R5 shows how the inversion symmetry of magnetization along the z -axis is broken by the Dzyaloshinskii-Moriya interaction D .

Fig. R5. Magnetic potential along the z -axis with and without the Dzyaloshinskii–Moriya interaction D . (a) In the case of $D=0$, the potential along the z -axis is a harmonic function with a minimum at $z=0$. (b) In the case of $D \neq 0$, the potential along the z -axis is an anharmonic function with a minimum at $z \neq 0$. For this graph, we used a value of D that is much larger than the actual value for HoFeO_3 .

Replies to Reviewer 4's comments

Comment 4-1

Zhang et al. have resubmitted their manuscript titled "Generation of third-harmonic spin oscillation from strong spin precession induced by terahertz magnetic near fields" and addressed the questions raised by the referees. In my case, my comments 4-2, 4-3, 4-5, 4-6 and 4-7 have been satisfactorily addressed. However, there is one problem remaining, which I address below.

Response 4-1

We thank Reviewer 4 for your clarifying his/her remaining concern, which we address in detail below. We are confident that also this problem can be addressed satisfactorily, and hope that the revised manuscript can be published in *Nature Communications*.

Comment 4-2

In my comment 4-4, I asked the authors to discuss the data of Fig. 1 for longer delay times. Indeed, as by my expectation, there is a revival of the oscillation amplitude at larger delay times, which the authors' figure in the supplementary material now shows. I am a bit puzzled that the authors did not comment on the possible origin of this potentially interesting feature.

In any event, it must be explained and discussed. It cannot result from the slightly inhomogeneous magnetic field distribution in the spiral antenna center, since the distribution is homogeneous (a beating feature as the observed one would require two distinct, separated spectral peaks and hence two distinct B field amplitude components). Since this oscillation represents, in my view, a potentially fundamental aspect of the physics presented here, it must be addressed before the manuscript can be further considered.

Response 4-2

We thank Reviewer 4 very much for pointing out that the origin of this feature at longer delay times was unclear. To clarify the origin of this feature, we added new calculation results to the Supplementary Information. Please confirm the details below.

As pointed out correctly by Reviewer 4, the magnetic field distribution near the center of the spiral antenna is homogeneous with respect to the x - y plane. On the other hand, the distribution with respect to the depth direction (z -axis) is highly inhomogeneous as shown in Fig. S4 and also in Fig. R6a below.

Since a magnetic field with a different amplitude excites a spin precession with a different red shift, the depth dependence of the magnetic field results in different precession motions that interfere when the probe pulse is transmitted through the sample.

A simple example is shown in Fig. R6b, where we consider the interference of $\Delta M_z/M_0$ components at four different depths. The considered z positions are marked by the arrows in Fig. R6a, and the corresponding magnetic field amplitudes are provided next to the arrows. As a result of the interference, a clear beating signal can be observed.

Fig. R6. Origin of the beating signal. (a) Depth dependence of the maximum peak amplitude of the magnetic near-field in HoFeO₃. The HoFeO₃ surface including the microstructure is located at $z = 0$. **(b)** The average of the $\Delta M_z/M_0$ components at the four z -positions marked in (a).

REVIEWERS' COMMENTS

Reviewer #1 (Remarks to the Author):

The authors add some additional results/discussions. All of the four referees evaluate the data quality positively, including me, though two referees question the novelty (compared to the requirement of previous submission to Nat. Photo.). I am happy to see the work becomes better after two iteration. I think the work is ready for the publication in Nat. Commun.

Reviewer #2 (Remarks to the Author):

Authors have revised the manuscript satisfactorily. By the way, I have one additional comment about the symmetry argument. In the Section XVI of the Supplementary Information, authors showed that the mirror symmetry is broken based on the potential energy along the z-axis. Would this imply that the inversion symmetry is broken as well? Also, it would be good to know the scale of the axes, in particular, x-axis. With an appropriate improvement of this part, I recommend the publication of this work in Nature Communications.

Reviewer #3 (Remarks to the Author):

Zhang et al. have resubmitted their manuscript, originally intended for Nature Photonics, to Nature Communications. My previous impression of the work was already very positive and the authors have now significantly improved their manuscript and in particular responded satisfactorily to my concerns.

I think the paper is now in very good shape and I have no reservations for publication in Nature Communications.

Final revisions for manuscript NCOMMS-22-51361

Title: Generation of third-harmonic spin oscillation from strong spin precession induced by terahertz magnetic near fields

Author: Z.-Y. Zhang et al.

General response to the recommendations and criticisms:

We would like to thank all reviewers for carefully studying our manuscript and our responses to their comments in the previous round. We also thank the Editor for providing the additional note regarding our previous response to Reviewer #2.

Regarding the final revisions, we replied to all comments and made minor revisions to our manuscript as explained below. The revisions are also highlighted in blue in the attached file (Final_revisions_Main_NCOMMS-22-51361_submitted.docx) for your convenience.

Figure updates

<Main text>

Figure 2: We modified the length of big vertical arrow for avoiding its overlaying with the data without any changes of content.

Figure 4: We modified the symbols of data without any changes of content.

<Supplementary Information>

Figure S7: We modified the symbols of data for the clarity without any changes of content.

Figure S10: We modified the symbols (solid line to dashed one) of data for the clarity without any changes of content.

Figure S11: We modified the symbols (circle, square, triangle) of data for the clarity without any changes of content.

Figure S12: We modified the symbols (solid line to dashed one) of data for the clarity without any changes of content.

Figure S13: We modified the symbols (circle, square, triangle) of data for the clarity without any changes of content.

Figure S18: We modified the label of x-axis (z to Z_1) without any changes of content.

Replies to Reviewer 1's comments

Comment 1-1

The authors add some additional results/discussions. All of the four referees evaluate the data quality positively, including me, though two referees question the novelty (compared to the requirement of previous submission to Nat. Photo.). I am happy to see the work becomes better after two iteration. I think the work is ready for the publication in Nat. Commun.

Response 1-1

We thank Reviewer 1 for acknowledging that the work has improved. We hope that the results can be published soon in *Nature Communications*.

Replies to Reviewer 2's comments

Comment 2-1

Authors have revised the manuscript satisfactorily. By the way, I have one additional comment about the symmetry argument. In the Section XVI of the Supplementary Information, authors showed that the mirror symmetry is broken based on the potential energy along the z-axis. Would this imply that the inversion symmetry is broken as well? Also, it would be good to know the scale of the axes, in particular, x-axis. With an appropriate improvement of this part, I recommend the publication of this work in Nature Communications.

Response 2-1

We thank Reviewer 2 for this comment about Fig. S18. Before we discuss the inversion-symmetry properties, we would like to briefly explain the meaning of the x-axis: The x-axis in this figure is modified and dimensionless, because Z_1 is the projection of the unit vector of the sublattice magnetization. As reviewer pointed out, we can discuss the mirror or inversion symmetries using the magnetic potential as described by Eq. (S6.2) in SI, because of considering the one-dimensional axis Z_1 . Thus, Fig. S18 shows that when the system possesses a finite DM interaction D , the inversion symmetry $Z_{1,2} \rightarrow -Z_{1,2}$ (or the symmetry of π rotation in Sec. XV), is broken, i.e., $V(Z_{1,2}) \neq V(-Z_{1,2})$. We modified the axes in Fig. S18 is also improved so that the magnetic potential is shown as a function of Z_1 , and added this information to the caption. The potential V is plotted in arbitrary units, because we use a value of D that is much larger than the actual value for HoFeO_3 .

Figure R1. Magnetic potential along the z -axis with and without the Dzyaloshinskii–Moriya interaction D . (a) In the case of $D=0$, the potential along the z -axis is a harmonic function with a minimum at $z=0$. (b) In the case of $D \neq 0$, the potential along the z -axis is an anharmonic function with a minimum at $z \neq 0$. For this graph, we used a value of D that is much larger than the actual value for HoFeO_3 . Note that “a.u.” is the abbreviation for “arbitrary units”, and the parameter Z_1 is dimensionless.

Replies to Reviewer 3's comments

Comment 3-1

Zhang et al. have resubmitted their manuscript, originally intended for Nature Photonics, to Nature Communications. My previous impression of the work was already very positive and the authors have now significantly improved their manuscript and in particular responded satisfactorily to my concerns. I think the paper is now in very good shape and I have no reservations for publication in Nature Communications.

Response 3-1

We thank Reviewer 3 for acknowledging that the manuscript has improved significantly. We hope that our results can be published soon in *Nature Communications*.